# AI Course Design Planning Framework: Developing Domain-Specific AI Education Courses

**Johannes Schleiss** [1,*] , **Matthias Carl Laupichler** [2] , **Tobias Raupach** [2] **and Sebastian Stober** [1]

[1] AG Artificial Intelligence Lab, Otto von Guericke University Magdeburg, 39106 Magdeburg, Germany; stober@ovgu.de

[2] Institute of Medical Education, University Hospital Bonn, 53127 Bonn, Germany; matthias.laupichler@ukbonn.de (M.C.L.); tobias.raupach@ukbonn.de (T.R.)

[*] Correspondence: johannes.schleiss@ovgu.de

**Abstract:** The use of artificial intelligence (AI) is becoming increasingly important in various domains, making education about AI a necessity. The interdisciplinary nature of AI and the relevance of AI in various fields require that university instructors and course developers integrate AI topics into the classroom and create so-called domain-specific AI courses. In this paper, we introduce the "AI Course Design Planning Framework" as a course planning framework to structure the development of domain-specific AI courses at the university level. The tool evolves non-specific course planning frameworks to address the context of domain-specific AI education. Following a design-based research approach, we evaluated a first prototype of the tool with instructors in the field of AI education who are developing domain-specific courses in this area. The results of our evaluation indicate that the tool allows instructors to create domain-specific AI courses in an efficient and comprehensible way. In general, instructors rated the tool as useful and user-friendly and made recommendations to improve its usability. Future research will focus on testing the application of the tool for domain-specific AI course developments in different domain contexts and examine the influence of using the tool on AI course quality and learning outcomes.

**Keywords:** AI education; AI teaching; AI literacy; course development tool; course planning

## 1. Introduction

Artificial intelligence (AI) literacy describes the broad, general knowledge and skills of individuals who interact with AI technology [1]. At the same time, the application of AI and the respective required competencies differ across domains and disciplines [2]. In this context, education about AI (AI education) goes beyond the concept of AI literacy and describes the education about domain-specific and interdisciplinary AI competencies that take into account the domain requirements and the application of AI in the domain. In this paper, we refer to this interdisciplinary education concept as domain-specific AI education.

While there are already several frameworks and experiences with AI literacy courses for different target groups [3–7], there is still a lack of domain-specific AI education courses. The interdisciplinary nature of domain-specific AI education poses a set of challenges. First, an understanding of the background, prior experiences and initial competencies of learners is necessary [8,9]. In the context of domain-specific AI education, students come from different disciplines and with different prior knowledge and skills, which are essential for a deeper understanding of AI [6]. This includes, for example, mathematical and statistical competencies, technical understanding and experience in computational thinking. Second, instructors teaching domain-specific AI courses often combine their experience in their domains with the additional cross-disciplinary topic of AI [10–12]. Thus, teaching a domain-specific AI course requires a thorough (self-)reflection of the instructors' competencies in AI and their role in the learning process [11]. Third, the complexities of AI are hard to grasp and the development of new AI technology and tools advances

at a rapid pace. At the same time, AI technology has different use cases, implications and underlying data in each discipline [13,14]. Thus, domain-specific AI education needs to bridge the understanding of AI and the understanding of requirements of its application in the domain.

To overcome the challenges faced by instructors and course developers, we propose a structured approach to designing domain-specific AI courses and help instructors to bring AI education into their respective domain. The proposed approach, the "AI Course Design Planning Framework", combines general course planning frameworks with AI- and domain-specific components. Therefore, it serves as a starting point for introducing AI to students in the context of their domain. The framework is designed to act as a guide through all questions that arise in the process of the course development for domain-specific AI courses and facilitates discussions among course developers, instructional designers and instructors to identify potential weaknesses or bottlenecks before starting the development of teaching materials and the actual course. It aims to overcome the identified challenges in developing domain-specific AI courses including understanding the background, prior experiences and initial competencies of learners, understanding the instructors competencies and most importantly, identifying the relevant learning outcomes for the course. Thus, the framework serves as an analyzing and rapid design tool that can support the alignment between stakeholders and the iterative development of courses.

The framework is developed and improved by using an iterative design-based research approach [15,16]. In design-based research, testing the usability of a particular (pedagogical) tool is often conducted as the first step in the process [17]. Only when it is clear that a tool is usable and useful, the actual effectiveness of certain design elements is tested in several iterations. Therefore, this preliminary study evaluated the two essential constructs of usability and usefulness to subsequently improve the applicability of the course planning framework in the course development process for domain-specific AI courses. This builds the necessary foundation for real-world implementation and evaluation of the framework in course design in different educational environments and in different domains. However, the implementation aspect is outside the scope of this paper as it focuses solely on evaluating usability and user experience. Thus, in addition to introducing the framework, this paper focuses on two research questions (RQ), namely:

- RQ1: How do instructors perceive the usability and user experience of the course planning framework as an instrument to structure and develop domain-specific AI courses?
- RQ2: What aspects of the framework could be improved to enhance its usability and user experience?

The paper is structured as follows. Section 2 introduces related research on AI education, interdisciplinary teaching and general course planning frameworks. Section 3 presents the AI Course Design Planning Framework. Section 4 describes the design-based research methodology which includes gathering data to evaluate the usability and usefulness of the framework. Results of the evaluation are presented in Section 5 and potential further improvements are discussed in Section 6. Section 7 concludes with a discussion of the most relevant findings and an outlook on further steps in domain-specific AI education.

## 2. Related Work

### 2.1. AI Education

In the context of this paper, AI education is defined as teaching AI as a subject ("teaching about AI") rather than including AI applications as tools in education ("AI in education and teaching") [18]. AI education can be distinguished into three areas that target different goals. First, AI literacy education aims at increasing the basic AI knowledge and skills of the general population. In this area, a wide variety of initiatives and experiences already exist that focus on different target groups, such as schoolchildren [7,19], college students [20,21], university students [5] or the general population [4]. AI literacy education can take many forms, from traditional classroom teaching to (open) online courses [22,23]. Materials for

AI literacy education often try to convey a basic understanding of AI concepts and provide accessible explanations about the potential, opportunities and risks of AI. Second, expert AI education focuses the competencies to further develop AI methods and AI research. It is aimed at a deep understanding of the theoretical foundations, modeling techniques, model architectures, current limitations in the field and possible advancements of methods [24]. Examples can be advanced graduate or post-graduate AI courses or research seminars, which often build upon established textbooks such as Russell and Norvig [25] and Goodfellow et al. [26]. Third, domain-specific AI education aims at the use of AI as a tool in the perspective and context of a professional or academic domain. Some examples for domain-specific AI education can be found in business education [27], teacher education [28] and in professional education [29]. However, it is becoming increasingly clear that students outside of computer science do not feel adequately prepared for the increased integration of AI into their field [30,31].

### 2.2. Interdisciplinary Teaching in Higher Education

Interdisciplinary teaching refers to synthesizing content and concepts from different disciplines into one teaching approach [8,9,32]. Van den Beemt et al. [9] surveyed recent approaches to interdisciplinary engineering education and found that these are often built with a vision to account for the complex real-world problem-solving, the social awareness of engineers, the entrepreneurial competencies as well as to improve disciplinary programs. Moreover, their study showed that interdisciplinary teaching approaches require careful consideration concerning student participation, the composition of learners, pedagogical approaches and assessments. Furthermore, interdisciplinary teaching needs to consider support structures for instructors and students.

These findings are supported by Lindvig and Ulriksen [8], who found that the justifications for interdisciplinary teaching mostly lay in developing (1) particular competencies, (2) interdisciplinary collaboration and (3) transferable competencies, such as creativity or communication. Moreover, Lindvig and Ulriksen [8] stress that interdisciplinary teaching activities can increase the motivation of students. Their analysis showed that interdisciplinary teaching activities take different forms but lean toward active and collaborative learning activities such as group work, case-based teaching, project-based work and problem-based learning. At the same time, one-third of the analyzed studies employ a lecture-based approach.

In the case of AI education, interdisciplinary teaching approaches are discussed from multiple perspectives. Janssen et al. [33] reported on experiences of an interdisciplinary AI master program. They built in six core characteristics in their course work: (1) courses are taught by multidisciplinary and interdisciplinary staff; (2) engineering techniques and theory are used hand-in-hand, connecting implementation to theoretical concepts; (3) students are given choices in assessment and presentations to allow for individual interests; (4) highlighting relevance to practice and industry; (5) highlighting multidisciplinary origins of machine learning; and (6) balancing skill levels.

Similar to point (5), Mishra and Siy [34] argued that AI teaching should also include the roots of the field from philosophy, neuroscience, psychology, cognitive science and others. Moreover, they distinguished between a computer-science-centric approach to AI courses and an interdisciplinary approach to AI courses that includes strong connections to related research fields.

Kong et al. [35] evaluated AI literacy courses for students with different non-expert backgrounds. They evaluated two courses that aimed at building a conceptual understanding of AI and found that participants felt empowered by the gains in their AI literacy and conceptual understanding. Moreover, they argued that the courses lower the entry barrier for AI literacy. Similarly, Ng et al. [23] discussed AI literacy courses for students from non-engineering backgrounds and argued that AI literacy should not be seen as a specialized field under engineering but should be seen as a competence for students from all disciplines and levels.



*2.3. Course Planning Frameworks*

Several general course planning frameworks and instructional design methodologies already exist. Examples are the ADDIE instructional design approach [36], Kern's six-step approach to curriculum development [37], Understanding by Design [38], Constructive Alignment [39] and the Merrill's Principles of Instruction [40].

The ADDIE approach [36] is an established process-based approach to instructional design, consisting of five development stages of analyzing, designing, developing, implementing and evaluating. It is a general, flexible, iterative and integrative learning design approach that helps educators identify necessary learning needs and develop appropriate learning activities to achieve desired learning outcomes.

Similarly, Kern's six-step approach to curriculum development [37] uses a generic, flexible and structured approach to plan and develop curricula and courses. The approach includes six steps, namely (1) problem identification and general needs assessment, (2) targeted needs assessment, (3) goals and objectives, (4) educational strategies, (5) implementation and (6) evaluation and feedback.

Understanding by Design is a course design approach that relies on the idea of backward design, starting with developing the learning outcomes of the course, the assessment and then the learning activities. It is closely connected to the Constructive Alignment approach [39], which argues for aligning learning outcomes, assessment methods and learning activities.

Focusing on the implementation of learning activities, Merrill's Principles of Instruction [40] established five instructional design principles for developing courses. According to the principles, learning is promoted when (1) learners engage in solving real-world problems, (2) existing knowledge is activated as a foundation for new knowledge, (3) new knowledge is demonstrated to the learner, (4) new knowledge is applied by the learner and (5) the new knowledge is integrated into the learner's world.

The idea of using a design tool as a practical and visual framework for lesson planning [41,42], lesson redesign [43] or curriculum development [44] has been tested before. The use of design tools is mostly inspired by ideas from Design Thinking [45] and the Business Model Canvas introduced by Osterwalder et al. [46]. The experiences working with these tools indicate that having a simple, concise visual framework that summarizes core ideas on a single page is valuable. Through their simple and visual structure, these design tools allow for rapid, iterative development and create alignment between stakeholders in the design process [41,42,46].

From the lenses of the ADDIE approach, the AI Course Design Planning Framework can be seen as an additional tool in the analyzing and design stages of the process that aims to get an overview of the course outcomes and structure. Similarly, it can be positioned as a structured approach for the first three steps of Kern's six-step approach. Following the ideas of Understanding by Design and Constructive Alignment, the AI Course Design Planning Framework aims to support educators to develop their respective learning outcomes in domain-specific AI courses. Most course planning frameworks build on identifying relevant learning outcomes based on the experience and pre-existing knowledge of the learners. Similar to other interdisciplinary course settings, the learning outcomes in domain-specific AI education are also influenced by the respective domain. Thus, when integrating AI education into the disciplines, course developers or instructors need to specify the application areas and implications of AI in the respective domain before looking at learning outcomes. Moreover, the requirements of learners and instructors with respect to AI experience and interaction need to be taken into account. To support course developers or instructors, we extend the general idea of course planning frameworks to identify relevant learning outcomes based on experience and pre-existing knowledge of learners towards specific considerations regarding AI in the domain. As such, we propose a concise planning framework for domain-specific AI course design.

## 3. AI Course Design Planning Framework

Figure 1 shows a graphical presentation of the AI Course Design Planning Framework as a concise course development tool (A blank version of the canvas is available for download under https://education4ai.github.io/ai-course-design-planning-framework/ (accessed on 23 August 2023)). The framework consists of three interacting "pillars", namely "AI in the domain", "learning environment" and "course implementation". The first pillar "AI in the domain" focuses on the external context of the application of AI in the domain. The second pillar reflects the learning environment in which the course takes place, such as learners and their interaction with AI, the competencies of the instructor and the available internal support. The third pillar describes the course implementation with learning outcomes, assessment and learning activities, all supported by the findings from the previous two pillars. Thus, the first two pillars have a supportive function. They can be interpreted in terms of Kern's six-step approach as the "needs assessment" component, which serves as the basis for the pedagogical structure of the course. In the following, we describe the framework from left to right and explain its intended use in the context of domain-specific AI education.

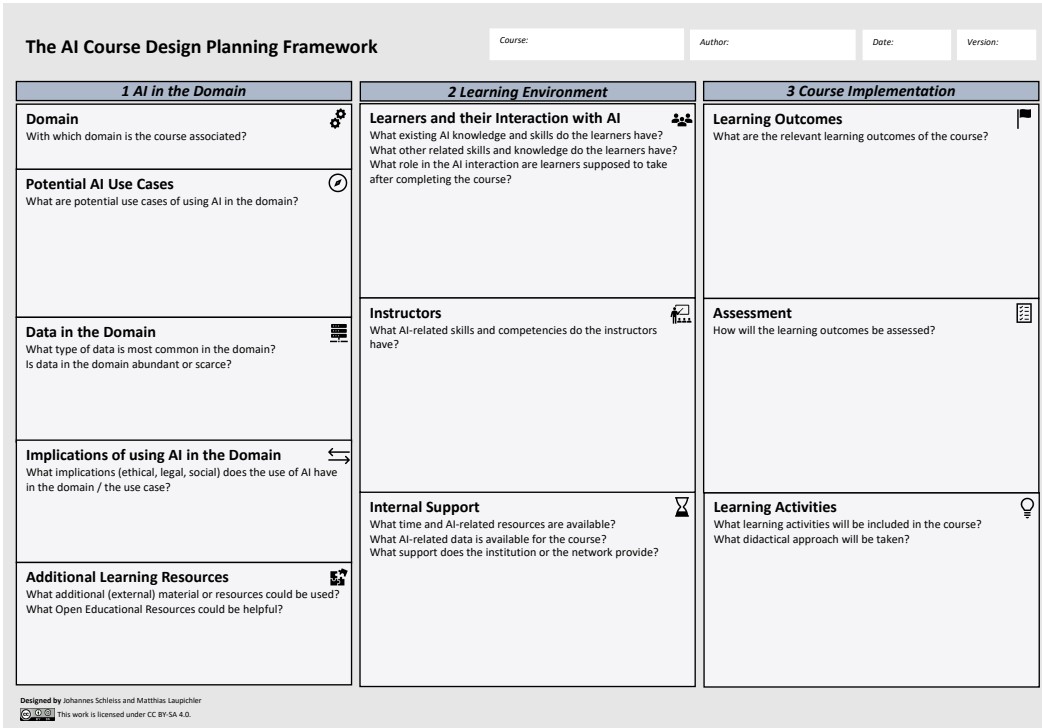

**Figure 1.** The AI Course Design Planning Framework with its three pillars focusing on (1) AI in the domain, (2) the learning environment and (3) the course implementation.

### 3.1. AI in the Domain

Describing the use of AI in the domain is the starting point of any endeavor to create a domain-specific AI course, as it determines what content will be taught in the corresponding courses. The subtopics of the pillar are presented in the following subsections.

### 3.1.1. Domain

The term domain is used to refer to the discipline in which AI is to be applied. For AI applications in medicine, an exemplary domain could be "radiology" and for AI applications in vehicle technology and development, it could be "mechanical engineering".

### 3.1.2. Potential AI Use Cases in the Domain

This subtopic is focused on the effect of AI technology in the domain. It helps to structure the topics that will become relevant for students and learners in the near future. Its goal is to support the identification of current use cases and the prognostic assessment of possible future use cases in which AI could play a role in the domain-specific problem solution.

### 3.1.3. Data in the Domain

The identified AI use cases are usually based on the most relevant type of data in the respective domain. The respective subtopic is not about which data is easy to obtain or how it can be used but rather about the type of data that is involved. Knowledge of typical data in a given domain enables more targeted use of AI techniques and specification of the data. It makes a big difference for the AI techniques which are to be taught whether the domain mainly works with time-series data, texts, images or other data types. Moreover, it is an important consideration whether the data in the domain is abundant or scarce.

### 3.1.4. Implications of Using AI in the Domain

Another important factor to consider is the potential implications that could arise when using AI in the respective field [47]. This mostly concerns ethical, legal and social implications [48,49]. For example, using AI to support medical triage decisions has different implications than using it to optimize the energy consumption of a manufacturing plant. Understanding the impact of technology on their domain helps students adhere to societal and ethical standards when using or developing AI technologies in their domain.

### 3.1.5. Additional Learning Resources

The creation of course material can be supported and guided by existing learning material and from oneself, colleagues or other institutions. In particular, Open Educational Resources (OER) can provide a basis for course development and can be used as preparatory or supplementary materials in course design [50,51].

Overall, the answers to the questions from this pillar build the foundation on which skills and knowledge are to be taught in the course units.

### *3.2. Learning Environment*

In addition to the external aspects of AI in the domain (Section 3.1), there are several ways in which the learning environment in which the course takes place can influence the pedagogical implementation of the course. Domain-specific AI courses as interdisciplinary courses place a special demand on learners, instructors and internal support. Thus, it is important to fully understand who the learners are, what skills the instructor possesses and which additional internal support is available.

### 3.2.1. Learners and Their Interaction with AI

Concerning the learners, three considerations are important for domain-specific AI courses. First, it is important to understand which AI skills and related competencies such as mathematical foundations, computational literacy, data literacy or programming skills the learners have acquired beforehand. Second, it is important to clarify the role of the group of learners regarding their interaction with AI to choose relevant demonstrations of AI-applications and an appropriate level of difficulty. The role can be described in different ways, e.g., using the taxonomy suggested by Faruqe et al. [52], which presents four groups whose frequency of contact with AI and AI competency requirements differ from each other. According to the authors, the levels in ascending order are "Consumers, the General Public and Policymakers", "Co-Workers and Users of AI Products", "Collaborators and AI Implementers" and "Creators of AI" [52]. Third, the existing competencies and the future role are influenced by the curricular integration of the course in an overall program. Moreover, the curricular integration determines if it is a mandatory course and

correspondingly the expected number of students in the course. Note, that depending on the interdisciplinarity of the course, the group of learners may be more heterogeneous with students from different fields and with different experiences.

### 3.2.2. Instructors

Next to learners, instructors play an important role in the learning process [53]. Domain-specific AI teaching requires a mix of sufficient AI knowledge, domain expertise and pedagogical skills to teach an interdisciplinary course as well as the motivation and time from an instructor's perspective. The AI knowledge of faculty and instructors tends to be quite heterogeneous, ranging from no previous AI experience to decades of AI research experience [11]. Thus, it is important to understand and assess the instructor's abilities to teach the course. However, if instructors self-assess their knowledge and skills in AI, attention must be paid to possible cognitive biases or heuristics, leading to an under- or overestimation of actual AI skills. In various other professional contexts, it has been found that people relatively rarely assess their skills correctly and it can be assumed that it is no different when assessing one's own AI knowledge and skills [54,55].

### 3.2.3. Internal Support

Internal support, such as budget, personnel restraints, the maximal duration of courses, available data, software and hardware, can be viewed as resources in a positive sense or as limitations in a negative sense. In the context of AI teaching, two important considerations are the availability of data and the availability of hardware and computing resources. Moreover, instructor support (e.g., through training), institutional barriers concerning interdisciplinary teaching and student support (e.g., through additional resources and infrastructure) play a role in designing an interdisciplinary course [9].

### *3.3. Course Implementation*

The right pillar represents the core of the framework as it combines the findings from the previous pillars. It aims to create a pedagogical structure that can be interpreted as a short version of the final course implementation. The pillar is structured following the Constructive Alignment approach [39], aligning the desired learning outcomes, the assessment of those outcomes and the respective learning activities.

### 3.3.1. Learning Outcomes

Defining the content and scope of the learning outcomes is an important building block in the context of domain-specific AI teaching that is informed by the considerations of the other pillars and determines the focus of the course. To organize learning outcomes in a structured, consistent and verifiable manner, it is recommended to formulate learning objectives [39]. Learning objectives should be specific, measurable, achievable, reasonable and time-bound (i.e., "SMART" objectives; [56]) wherever possible. Furthermore, they should focus on specific competence levels following Bloom's Taxonomy [57]. The course learning objectives determine the structure of the course and indicate the time and resources spent on the individual topics. The learning objectives should be shared with the students so that they know which aspects of the course are the most relevant for their professional development.

### 3.3.2. Assessment

Following the Constructive Alignment approach, it is important to consider in advance through which methods and in which way the fulfillment of the learning objectives will be evaluated [39]. Assessment in interdisciplinary courses requires to balance the experiences of different groups of learners as well as the targeted outcome with respect to their interaction with AI. In addition to traditional assessment methods such as exams, tests oral presentations or reports, the applied nature of domain-specific AI teaching can also benefit from project- or problem-based assessments that are connected to real-world applications

(see [58] as an example). Moreover, research in interdisciplinary education indicates that using assessment through reflection can help students to bridge the disciplinary silos [9]. Similar to other fields, using different assessment components can be a beneficial and fair approach to account for the different experiences of students from different disciplines [33].

### 3.3.3. Learning Activities

The last step focuses on the learning activities that lead to the desired learning objectives [39]. Thus, the focus is on the pedagogical implementation of the overall course design. In this context the Merrill principles of learning [40] should be considered to promote an effective learning experience. Experiences from the few domain-specific AI courses that are already conducted today, indicate that a combination of different teaching methods is often used to address the different aspects of AI [27]. The overview of the learning activities builds the basis for more detailed planning of the learning activities throughout the course. These could also include using AI-based learning activities.

### 3.4. Intended Use of the AI Course Design Planning Framework

After describing the pillars of the framework and their underlying categories, we briefly explain the intended use in the context of course development for domain-specific AI courses. The AI Course Design Planning Framework forms a visual and practical tool for instructors and course developers in the higher education or professional education context with a special focus on non-computer science (non-CS) students. It can be used as means to gather ideas, innovate, plan and communicate ideas for domain-specific AI courses. The framework can be used as a self-contained instrument for individuals, in tandem with AI and domain experts or in a workshop setting with multiple people. We suggest filling it from left to right, first considering the questions on AI in the domain, the learning environment of the course and last the course implementation.

As a reference and further explanation, two filled examples for AI courses in mechanical engineering (Figure A1) and in radiology (Figure A2) can be found in the Appendix A.

## 4. Methods

### 4.1. Design-Based Research

This study represents the first iteration of a design-based research project [15,16]. The project aims to structure and facilitate the development of AI courses for non-CS students (i.e., students majoring in a field outside of computer science) and to identify potential bottlenecks for implementation. The central element of the project is the AI Course Design Planning Framework, which is to be continuously improved in accordance with the design-based research approach. The design-based research characteristics reported by Wang and Hannafin [59] are taken into account, with a special focus on pragmatism, theoretical support (i.e., grounded in relevant research) and iteration. This study is focused on the aspect of usability and user experience of the framework [17].

### 4.2. Procedure

A workshop was conducted in November 2022, which was specifically targeted at lecturers and other individuals involved in the development of AI courses for non-CS students. The invitation for the two-hour online workshop was shared on social media and promoted by a major AI-MOOC platform on its website and via its newsletter. No target group-specific advertising was placed, as it could be assumed that a sufficient number of representatives of the target group were addressed by the recruiting methods described. Workshop participation was free and voluntary and participants registered by filling out an online form. Since all workshop participants were German native speakers, the workshop language was German. Of 22 people who registered for the workshop, 18 actually attended. The first part of the workshop explained the concept of domain-specific AI education in more detail (refer to Section 2). In this context, we described the difference between general AI literacy and domain-specific AI education. Moreover, the importance of the latter for

today's students was discussed. In the second part, the participants were presented with the framework, which they were then asked to fill out in smaller groups. The original version of the framework used in the workshop can be found in the Appendix A, Figure A3. After filling each section, the findings were shared with the other groups. Following the workshop, participants were asked to complete an online questionnaire that included questions about the usability and user experience of working with the framework as well as open items to improve it. Following the iterative design-based research approach, the findings and the feedback of the participants was taken into account to further improve the underlying prototype.

### 4.2.1. Evaluation Instruments

Two well-known and widely used scales were applied to support the evaluation of the tool: The System Usability Scale (SUS) [60] and the User Experience Questionnaire (UEQ) [61]. In its original version, the SUS consists of 10 items and was initially developed for the rapid and cost-effective evaluation of industrial systems. Since its development, however, the SUS has been used in a variety of application domains [62] and has also been employed to evaluate a variety of educational technologies [63]. Because the proposed framework is not a technological system, three items were omitted (items 1, 4 and 5 from [60]). The UEQ items were presented in the same way as recommended by Laugwitz et al. [61]. In addition to the two scales, two open-ended questions were asked about traits of the framework that were particularly helpful and about aspects that could be improved.

### 4.2.2. Data Analysis

The data were analyzed using Microsoft Excel and the automatic data analysis of the survey program. The SUS was analyzed according to the recommendations made by Brooke et al. [60], who suggest that the scores of the negatively worded items be inverted first, then the average of each item be added. Ref. [60] recommends to multiply the sum by 2.5 to convert the scores from 0–40 to a composite measure of the overall usability between 0 and 100, which in our case means multiplying the sum by 3.571 to account for the smaller number of items. For the evaluation of the UEQ, the average of each item was determined, which are presented on a semantic differential (see Figure 2). The responses to the open-ended questions were analyzed qualitatively. Statements which appeared in a similar way in the answers of different workshop participants were paraphrased and are reported in the next section.

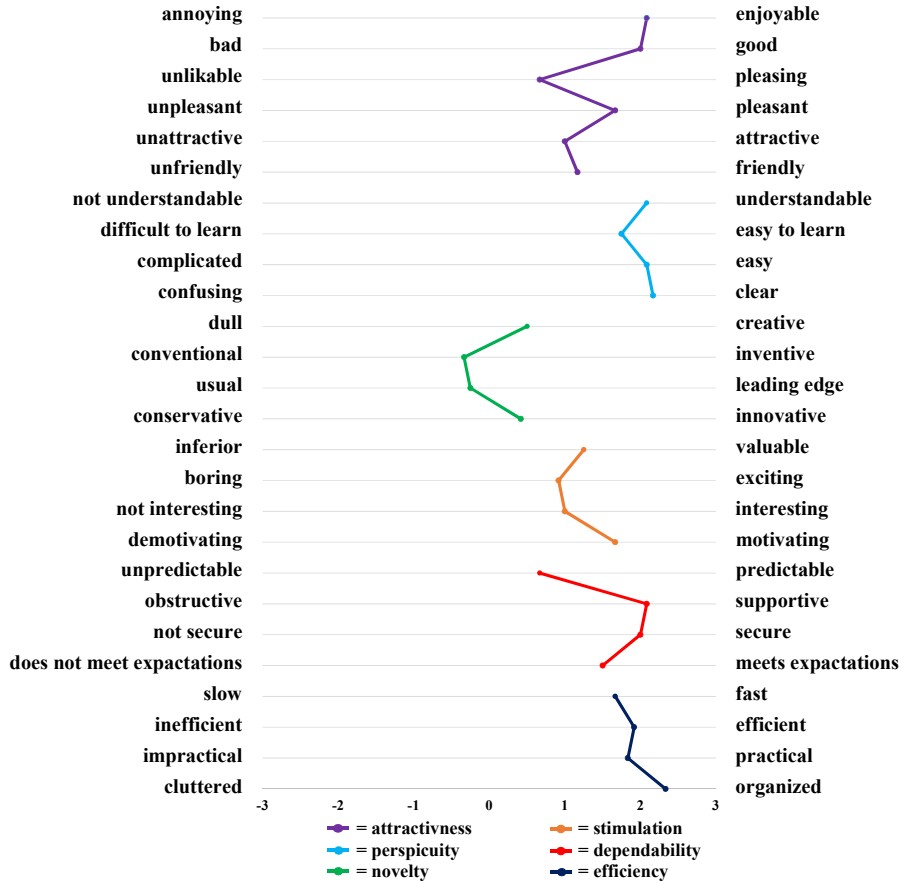

**Figure 2.** The average score on a scale of −3 to +3 for each UEQ item. The individual UEQ subcategories "attractiveness", "perspicuity", "novelty", "stimulation", "dependability" and "efficiency" are separated by color.

## 5. Results

### 5.1. Participants

Of the 18 workshop participants, 12 (66%) completed the questionnaire. To understand how the sample of participants reflect the generalizability of the findings, we assessed the background, occupation and self-reported AI expertise. Three participants stated that they came from the field of AI teaching, three were from the field of medicine, two were educational psychologists, two came from the field of organizational and university development and one person was connected to life sciences. When asked about their occupation, 6 participants (50%) indicated they worked at a university of applied sciences, 5 (42%) worked at a university and one person was from another educational institution. The level of AI expertise also varied widely among participants. While some participants had either already developed AI applications or conducted AI research themselves (n = 3, 25%) or had been involved with AI for a long time (n = 2, 16%), 4 participants (33%) reported having a good understanding of AI and 3 (25%) had only a rough idea of how AI works.

### 5.2. Usability (RQ1)

All participants completed all items from both the SUS and UEQ. All SUS items were recoded according to the instructions by [60]. After adding the average of all seven items, the sum was multiplied by 3.571 to obtain the final SUS score. The resulting items are presented in Table 1. The final score was 81.2, which according to the item benchmarks presented by Lewis et al. [64] corresponds to an "A" (grade system A+ to F) and is in the 90th to 95th percentile. Thus, concerning RQ1, the scores indicate a very good perceived usability with using the framework.

**Table 1.** Descriptive statistics for the SUS items with mean, standard deviation (SD) and range. Items with negative connotations are marked as (−).

| SUS-Item | Mean | SD | Range |
| --- | --- | --- | --- |
| I found the system unnecessarily complex. (−) | 2.0 | 1.1 | 4 |
| I thought the system was easy to use. | 4.3 | 0.4 | 1 |
| I thought there was too much inconsistency in this system. (−) | 1.9 | 0.8 | 2 |
| I would imagine that most people would learn to use this system very quickly. | 4.7 | 0.5 | 1 |
| I found the system very cumbersome to use. (−) | 1.8 | 0.9 | 3 |
| I felt very confident using the system. | 3.9 | 0.6 | 2 |
| I needed to learn a lot of things before I could get going with this system. (−) | 1.4 | 0.6 | 2 |

### 5.3. User Experience (RQ1)

On average, the participants rated the attractiveness of the AI Course Design Planning Framework with 1.43 on a scale of −3 to +3 (standard deviation (SD) = 0.89, 95% confidence interval (CI) [0.93, 1.94]), which can be interpreted as "above average" when compared to a benchmark [65]. Perspicuity was rated with 2.02 (SD = 0.80, 95% CI [1.57, 2.48]; "excellent"), efficiency with 1.94 (SD = 0.64, 95% CI [1.58, 2.30]; "excellent"), dependability with 1.56 (SD = 1.02, 95% CI [0.98, 2.14]; "good"), stimulation with 1.21 (SD = 1.10, 95% CI [0.59, 1.83]; "above average") and novelty with 0.08 (SD = 0.99, 95% CI [−0.48, 0.64]; "bad"). The average UEQ scores per item are shown in Figure 2.

### 5.4. Qualitative Responses (RQ2)

With respect to RQ2 of aspects to improve the framework, many of the workshop participants praised the structuring possibilities offered by the AI Course Design Planning Framework. For example, seven participants appreciated the ability to use the framework to conduct structured development and evaluation of AI courses. Moreover, three participants liked the questions in the individual text boxes that support the concretization of course development projects. One participant pointed out the possibility of using the framework to reflect on one's own (institutional) situation in relation to AI education. Beyond the support for concrete course development, one participant praises the ability to use the framework to organize and evaluate the more abstract development of entire AI curricula.

The open-ended, qualitative questions also asked for suggestions for improving the framework used in the workshop (Figure A3). In this context, four participants expressed a desire for more detailed explanations of the individual framework fields and their interrelation. In addition, two participants made recommendations on how the layout of the questionnaire could be improved for greater comprehensibility. One of the participants suggested that framework users should be provided tips for tools that might facilitate the application of the framework. Lastly, one participant requested that the potential transfer of internal factors be addressed more.

These considerations were used to improve the framework to the version presented in Section 3. Changes to the prototype based on the participants feedback are summarized Table A1.

## 6. Discussion

### 6.1. Discussion of the Results

We proposed an AI Course Design Planning Framework as a tool for developing domain-specific AI courses for non-CS students, which was evaluated by participants in an online workshop. By answering the first research question, we found that participants were generally positive about the course. This applies to both the usability of the course and the user experience. In particular, the user experience subcategories "perspicuity" and "efficiency" were rated very favorable by the workshop participants. This could indicate

that the framework is easy to understand and use and allows for a speedy and structured organization of the AI courses.

The only user experience subcategory that did not score particularly well was the "novelty" of the framework. When comparing the AI Course Design Planning Framework to the UEQ benchmark, which contains the results of various other projects that used the UEQ, the framework performed rather poorly in this subcategory. There may have been two main reasons for this: First, unlike many of the benchmark studies, the framework is not a new technological system, but a tool for developing AI courses. Second, the framework integrates existing models (as described above) and adds aspects to them that are essential for AI courses in particular. Accordingly, the goal of the framework was not to create a system that is as innovative as possible but to extend established approaches and specify them for AI.

The good usability rating was possibly due to the fact that the end-user was at the forefront of all considerations during the development of the framework. However, it must be mentioned again that only seven of the original ten SUS items were used since three items can only be used for the evaluation of technological systems. Thus, the interpretation of the SUS score in accordance with [64] must be carried out with some caution. As mentioned earlier, this study was the first step of an iterative design-research approach. Accordingly, as described in research question 2, the study aimed to find which aspects of the framework could be improved to optimize the usability and user experience of the framework. Therefore, the framework was further developed on the basis of the quantitative and particularly on the qualitative feedback (see Appendix A for the version of the framework used).

In the context of the free-text questions, several participants indicated that they would have preferred the columns to be numbered. In addition, some participants recommended that the columns be structured from left to right, rather than placing the "Pedagogical Structure" column in the center, as we did before. It was also mentioned that the questions should be defined more precisely and that some of the fields in the framework should have multiple questions. The suggested changes have been incorporated into the updated version of the framework, which is presented in Figure 1.

### 6.2. Limitations

One limitation of the study presented here is the small sample size, with only 12 subjects involved. Initially, this sample size may appear unusually small, potentially raising concerns about obtaining meaningful results. However, there are two crucial considerations to take into account when evaluating the sample size. First, it is important to recognize that developing AI courses for non-computer science students involves only a limited pool of individuals. Consequently, the 12 participants can be considered as a substantial representation of the total population of AI course developers within this context. Second, it is worth noting that the study's participants are experts in the field of AI course development, having engaged extensively in the creation of higher education (AI) courses in multiple domains. As such, their insights hold significant validity and can be considered valuable for the research. By considering these factors, the study's findings and conclusions gain credibility despite the relatively small sample size.

Another limitation pertains to language. The workshop language was German and the original framework was presented in German. Consequently, there is a possibility that translating the materials into English could potentially affect the study's validity. To address this concern, a validation study utilizing the English version of the framework is necessary.

Finally, this preliminary study only investigates the usability and user experience of the framework without examining the implementation of a course using the framework. While such an endeavor would add great value to the body of knowledge and is intended to be realized in future research projects, the implementation aspect was beyond the scope of this paper.

### 6.3. Strengths and Implications

The initial results indicate that the framework is both valuable and user-friendly. Its ease of use makes it well-suited for implementation in the complex realm of real-world course planning. By bridging the gap between traditional course development methods and innovative approaches that integrate AI into various disciplines, this research project contributes to the essential enhancement of AI education, preparing future professionals for active involvement in AI teams.

As mentioned previously, this study represents the initial step of an ongoing iterative design-based research project. Consequently, we have not yet tested whether utilizing the framework positively impacts the learning outcomes and quality of potential AI courses. Subsequent iterations will delve into this aspect, examining its influence on teaching quality in greater depth. Nevertheless, even at this stage, the AI Course Design Planning Framework serves as a valuable visual and practical tool, effectively structuring the development of new domain-specific AI courses.

## 7. Conclusions and Outlook

We introduced the AI Course Design Planning Framework to facilitate the development of domain-specific AI courses. The framework's application, user experience and usability were tested in the first design iteration involving 18 higher education course developers. The feedback indicated that the framework is user-friendly and valuable in supporting the creation of domain-specific AI courses.

Our future research will concentrate on two main aspects: further refining the framework and examining its application in diverse domain contexts. Additionally, we plan to conduct validation studies to assess how the framework enhances course quality in different domain settings.

Beyond the scope of this paper, we have identified several promising research directions that can advance the field of domain-specific AI education. First, establishing a learning catalog comprising AI-related competencies within specific domains and roles would be a valuable endeavor. This catalog could draw from existing frameworks, such as the European Skills/Competencies, Qualifications and Occupations (ESCO) [66] or other competence frameworks for AI in various domains [29,67,68]. Second, conducting more systematic research on pedagogical approaches for domain-specific AI education and interdisciplinary education would contribute significantly to the field's development. Third, exploring the role of integrating external materials, such as Open Educational Resources, into AI education efforts can cater the needs of instructors and address a wide range of learning profiles [29,50,51]. Finally, it would be beneficial to develop assessment instruments to measure general and domain-specific AI skills. These would support the evaluation of individuals' AI competency in their professional domain and beyond as well as allow us to build evidence around educational interventions [5,69].

**Author Contributions:** Conceptualization, J.S.; methodology, J.S. and M.C.L.; validation, J.S. and M.C.L.; formal analysis, M.C.L.; investigation, J.S. and M.C.L.; resources, T.R. and S.S.; data curation, J.S. and M.C.L.; writing—original draft preparation, J.S. and M.C.L.; writing—review and editing, J.S., M.C.L., T.R. and S.S.; visualization, J.S.; supervision, T.R. and S.S.; funding acquisition, S.S. All authors have read and agreed to the published version of the manuscript.

**Funding:** This research and APC was funded by German Federal Ministry of Education and Research, grant number 16DHBKI008.

**Institutional Review Board Statement:** Not applicable.

**Informed Consent Statement:** Informed consent was obtained from all subjects involved in the study.

**Data Availability Statement:** The data presented in this study are available on request from the corresponding author. The data are not publicly available due to privacy reasons.

**Acknowledgments:** We would like to thank Dana-Kristin Mah and Vivian van der Werf for her valuable feedback.

**Conflicts of Interest:** The authors declare no conflict of interest.

## Abbreviations

The following abbreviations are used in this manuscript:

| | |
|---|---|
| AI | Artificial Intelligence |
| CI | Confidence Interval |
| CS | Computer Science |
| ESCO | European Skills/Competencies, Qualifications and Occupations |
| MOOC | Massive Open Online Course |
| OER | Open Educational Resources |
| RQ | Research Question |
| SD | Standard Deviation |
| SMART | Specific, Measurable, Achievable, Relevant and Time-Bound |
| SUS | System Usability Scale |
| UEQ | User Experience Questionnaire |

## Appendix A

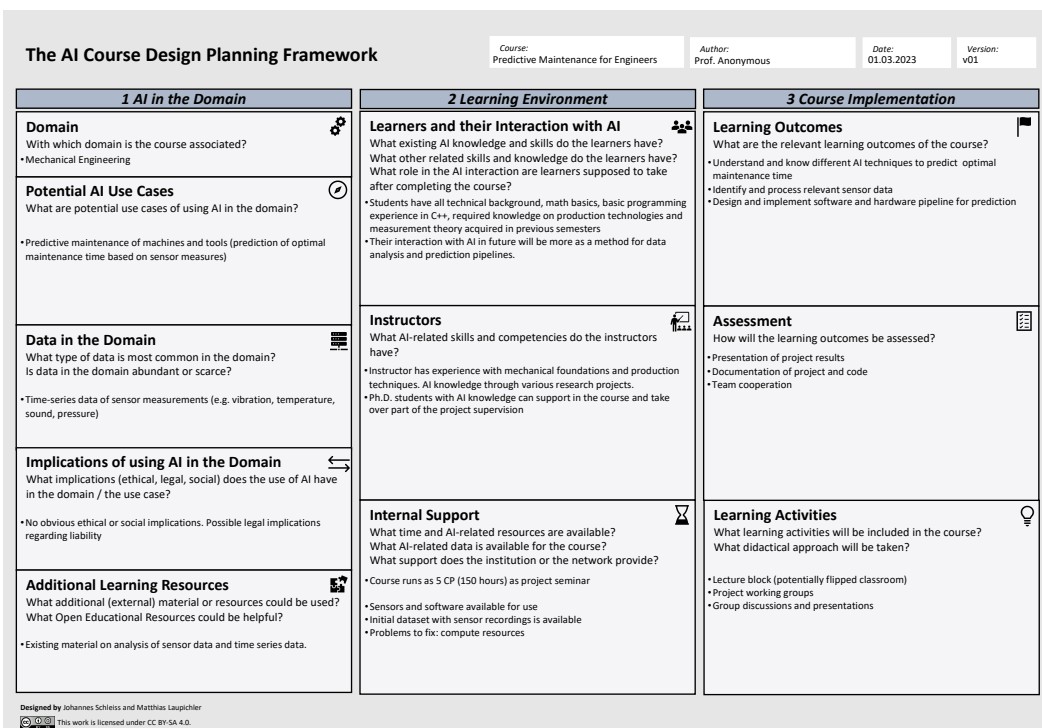

**Figure A1.** Exemplary AI Course Design Planning Framework filled for a course of predictive maintenance in engineering.

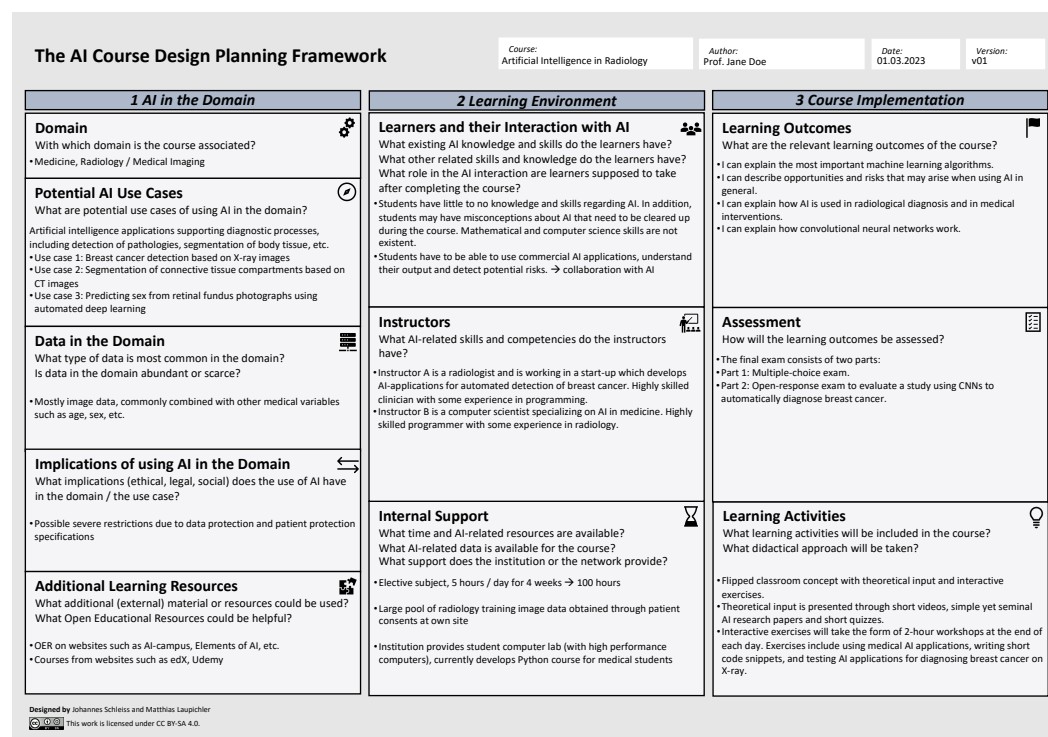

**Figure A2.** Exemplary AI Course Design Planning Framework filled for a course of radiology.

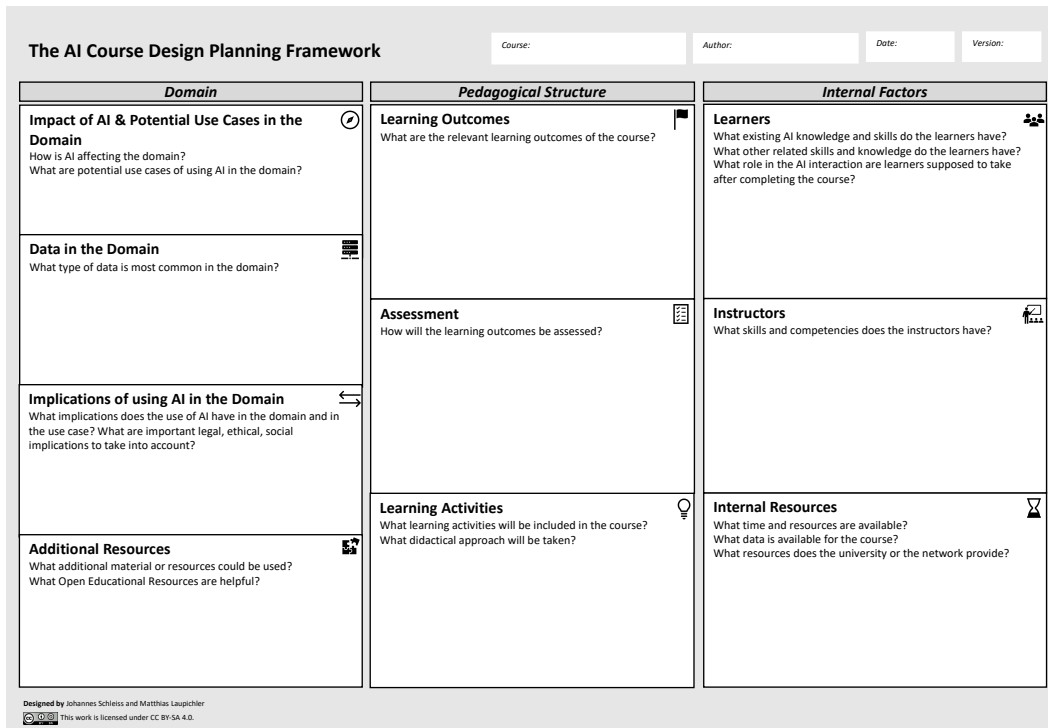

**Figure A3.** Original version of the framework used in workshop.

**Table A1.** List of modifications of the planning framework based on suggestions from experts using initial framework (Figure A3).

| Type of Change | Modification |
| --- | --- |
| Design and Layout | 1. Changing the order of pillars to reflect the order in which they should be filled<br>2. Numbering the pillars<br>3. Improving readability through different coloring |
| Clarity and Usability | 1. Renaming the pillars and the categories<br>2. Adding additional category of domain<br>3. Improving the clarity of the guiding questions in each category |

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
