# Peer review of "AI Course Design Planning Framework: Developing Domain-Specific AI Education Courses"

_education, doi:10.3390/educsci13090954_

Round 1

Reviewer 1 Report

In general, this paper exhibits commendable writing quality and introduces a framework with potential utility in the design of domain-specific artificial intelligence (AI) courses. The research assessing the usability and user experience of the framework offers preliminary evidence of its worth. The following suggestions could potentially enhance the overall quality and coherence of the paper:

The purpose of this study is to investigate the relationship between social media usage and mental health outcomes among adolescents. This topic is of great importance due to the increasing prevalence of social media platforms and the potential impact they may

The introduction may benefit from further elaboration regarding the primary obstacles and deficiencies that the framework seeks to tackle in the development of domain-specific artificial intelligence (AI) courses. This would serve to establish the necessity of the suggested framework.

The literature evaluation would benefit from a more comprehensive analysis of established frameworks pertaining to the creation of interdisciplinary or domain-specific courses. This analysis should include a thorough examination of how the proposed framework aligns with and diverges from previous research in this area.

The present study employed various methods to investigate the research question.

Enhancing this section would involve incorporating additional information regarding the workshop recruiting process and the backgrounds of the participants. For instance, it would be beneficial to include details on the methods employed to recruit participants, as well as their respective areas of expertise.

The rationale behind the authors' decision to prioritize the evaluation of usability and user experience as the initial phase, rather than focusing on actual implementation, could be elucidated by the authors.

Further rationale is required to support the utilization of a small sample size, as the inclusion of only 12 individuals appears to be somewhat inadequate. The inclusion of individuals characterized as "experts" in the discussion is a valuable contribution. However, it would be advantageous to provide additional justification for this choice.

The present article necessitates a comprehensive elucidation of the backgrounds of the participants. Given the limited sample size of merely 12 participants, it is imperative to ascertain the extent to which they can accurately reflect the broader community.

It is advisable for the author to include the outcomes of qualitative analysis.

The clarity of the presentation of the quantitative findings is evident. When analyzing the qualitative feedback, it is advisable to categorize the suggestions topically instead of merely presenting them as a list of impressions.

The present discourse revolves around the topic of discussion.

There is potential for further expansion of the restrictions section. The authors have the potential to address constraints pertaining to the limited sample size and the potential lack of generalizability of the findings.

Incorporating concrete illustrations of the framework's modifications in response to user feedback would serve to enhance the exposition of how the iterative design process facilitated enhancements.

The implications and future work sections provide valuable guidance for advancing this research.

In general, the present line of inquiry has promising potential, and the manuscript effectively adds to the existing body of knowledge. 

Reviewer 2 Report

Thanks to the authors for taking the initiative to design a framework for planning and developing AI courses for non-computer science students. The framework is well-organized, practical, and relevant, given the generative AI explosion that impacts all courses in higher education. The tool is helpful for me to use for integrating AI into the courses I teach.

The paper is interesting to read and well-written. I appreciate the careful editing and 
clearly-stated study design of this preliminary research. The authors acknowledge the study limitations of a small sample size while demonstrating how the participants contributed to meaningful results and credibility. 

I wonder if the AI Course Design Planning Framework should include a section for analyzing AI tools and chatbots that can be meaningfully integrated into the course activities and assignments.

The examples in Figures A1 and A2 are helpful and demonstrate alignment across the domain, learning environment, learning objectives, assessment, and activities. I believe that title of Figure A3 needs to be updated to "The original version of the framework used in the workshop."

In future studies involving the
AI Course Design Planning Framework, you may want to include more discussion and analysis around the need for developing AI literacy in both instructors and students. AI literacy is now essential for all in higher education. I look forward to learning more about how the AI Course Design Planning Framework affects course quality and teaching effectiveness in different contexts.

Overall, the authors make a relevant and original contribution to knowledge. The article is ready for publication.
